# Population-scale proteome variation in human induced pluripotent stem cells

**Bogdan Andrei Mirauta**[1†], **Daniel D Seaton**[1†§], **Dalila Bensaddek**[2†#], **Alejandro Brenes**[2], **Marc Jan Bonder**[1], **Helena Kilpinen**[1¶], **HipSci Consortium**[1,2,3,4], **Oliver Stegle**[1,5,6‡*], **Angus I Lamond**[2‡*]

[1]European Molecular Biology Laboratory, European Bioinformatics Institute, Wellcome Genome Campus, Hinxton, United Kingdom; [2]Centre for Gene Regulation & Expression, School of Life Sciences, University of Dundee, Dundee, United Kingdom; [3]Wellcome Sanger Institute, Hinxton, Cambridge, United Kingdom; [4]King's College, London, United Kingdom; [5]European Molecular Biology Laboratory, Genome Biology Unit, Heidelberg, Germany; [6]Division of Computational Genomics and Systems Genetic, German Cancer Research Center, Heidelberg, Germany

**\*For correspondence:**
oliver.stegle@embl.de (OS);
a.i.lamond@dundee.ac.uk (AIL)

[†]These authors also contributed equally to this work
[‡]These authors also contributed equally to this work

**Present address:** [§]Human Genetics, GSK R&D, Stevenage, SG1 2NY, United Kingdom; [#]Bioscience Core Laboratories, King Abdullah University of Science and Technology, Building 2, Office 2271, Thuwal, 23955-6900, Saudi Arabia; [¶]Helsinki Institute of Life Science, University of Helsinki, Helsinki, Finland

**Group author details:**
HipSci Consortium See page 17

**Competing interests:** The authors declare that no competing interests exist.

**Abstract** Human disease phenotypes are driven primarily by alterations in protein expression and/or function. To date, relatively little is known about the variability of the human proteome in populations and how this relates to variability in mRNA expression and to disease loci. Here, we present the first comprehensive proteomic analysis of human induced pluripotent stem cells (iPSC), a key cell type for disease modelling, analysing 202 iPSC lines derived from 151 donors, with integrated transcriptome and genomic sequence data from the same lines. We characterised the major genetic and non-genetic determinants of proteome variation across iPSC lines and assessed key regulatory mechanisms affecting variation in protein abundance. We identified 654 protein quantitative trait loci (pQTLs) in iPSCs, including disease-linked variants in protein-coding sequences and variants with *trans* regulatory effects. These include pQTL linked to GWAS variants that cannot be detected at the mRNA level, highlighting the utility of dissecting pQTL at peptide level resolution.

## Introduction

Induced pluripotent stem cells (iPSC) hold great promise for advancing basic research and biomedicine. By enabling the in vitro reconstitution of development and cell differentiation, iPS cells allow the investigation of mechanisms underlying development and the aetiology of many forms of genetic disease. To realize this potential, it is essential to characterise genetic and non-genetic sources of variability of molecular and cellular phenotypes in human iPSCs.

Recently, multiple reference panels of human iPSC lines have been established (*Kilpinen et al., 2017*; *Panopoulos et al., 2017*; *Carcamo-Orive et al., 2017*), providing a valuable resource for functional experiments in pluripotent cells. These cell lines, together with associated data, have enabled the characterisation of variability in iPSC transcriptomes, identifying genetic and non-genetic determinants of expression variation, including expression quantitative trait loci (eQTL) (*Kilpinen et al., 2017*; *Rouhani et al., 2014*; *DeBoever et al., 2017*) in cis.

While RNA-centric analyses are informative for studying gene regulatory mechanisms at the transcriptional level, most cellular phenotypes ultimately involve downstream mechanisms that are mediated by proteins. Several proteogenomics studies, primarily in cancer (*Zhang et al., 2014*; *Mertins et al., 2016*), have underlined the relevance of protein measurements to interpreting how genomic changes act at the phenotypic level. Moreover, recent evidence has shown that genetic

alterations can have effects on RNA that are attenuated at the protein level (*Gonçalves et al., 2017*; *Roumeliotis et al., 2017*). Vice versa, the mapping of protein quantitative trait loci (pQTL), predominantly in lymphoblast cell lines (*Battle et al., 2015*; *Stark et al., 2014*; *Wu et al., 2013*) and for the plasma proteome (*Sun et al., 2018*; *Yao et al., 2018*; *Liu et al., 2015*; *Johansson et al., 2013*; *Lourdusamy et al., 2012*), has revealed genetic effects on protein traits that do not manifest at the RNA level. However, the extent of RNA-independent protein regulation is not yet understood, with previous analyses performed only at gene-level resolution and, in some cases, without comparing protein and RNA data from the same cellular material.

Here, we report the first comprehensive, population-scale, combined proteomic and transcriptomic analysis of human iPSC lines. Our data provide matched quantitative proteomic (Tandem Mass Tag Mass Spectrometry) and transcriptomic (RNA-Seq) profiles of 202 iPSC lines, derived from 151 donors from the HipSci project (*Kilpinen et al., 2017*). We identify both genetic and non-genetic effects associated with variability in protein expression between individuals and describe the first high-resolution pQTL map in human iPSCs, including loci not detected as eQTLs at the RNA level.

## Results

### A population reference proteome for human iPSCs

A set of 217 iPSC lines from the HipSci project (*Kilpinen et al., 2017*), derived from 163 distinct donors, was selected for protein analysis, using material from the identical batches of cells that were used for RNA-Seq and other assays (Materials and methods). Quantitative mass spectrometry was carried out in batches of 10 lines, using tandem mass tagging (TMT, *Thompson et al., 2003*), with one common reference sample shared across batches (*Brenes et al., 2018*) (Materials and methods). Collectively, we identified 255,015 distinct (unmodified) peptide sequences, corresponding to 16,773 protein groups (groups of protein isoforms with no discriminating peptides; hereon denoted proteins) with median sequence coverage of 46%.

After quality control, 202 lines (from 151 donors) with matched genotype, RNA-Seq and proteome data, were selected for further analysis (*Figure 1a*; *Figure 1—figure supplement 1*; *Figure 1—source data 1*). We identified 11,140 recurrently detected proteins, corresponding to 10,198 genes (detected in at least 30 lines; Materials and methods) and RNA expression for 12,363 protein-coding genes (population average TPM >1). Out of these, 9013 protein coding genes were detected both at the RNA and protein levels (*Figure 1—source datas 2* and *3*).

Collectively, these data provide the most comprehensive analysis of the human iPSC proteome reported to date, and one of the most comprehensive proteomic datasets reported for any human primary, or derived, cell type (*Supplementary file 1*). When overlaying our data with the Human Proteome Map (*Kim et al., 2014*), the iPSC proteome was most similar to foetal and reproductive organs (*Figure 1—figure supplement 2*), which is consistent with the expected expression of pluripotency markers in these tissues (*Kerr et al., 2008a*; *Kerr et al., 2008b*). We also assessed differences between healthy donors and disease-bearing donors, identifying no systematic expression differences in the iPSC state (*Supplementary file 2*; *Figure 1—figure supplement 3*).

### RNA and proteome variability

We assessed a range of factors to explain the variation in protein expression between iPSC lines. Leveraging our experimental design with data from two or more lines for 34% of donors, we assessed the effects of donor, alongside age, sex, and the contributions of technical and cell culture related factors (on 6009 genes; *Figure 1b,d*; *Figure 1—source data 4*; using a linear mixed model; Materials and methods). Overall, the fraction of variance explained by biological factors was lower for protein levels, compared to RNA variation, which points to higher assay noise and/or stochastic variability of protein abundance. Consistent with previous results using RNA data (*Kilpinen et al., 2017*), we identified donor genome (i.e. DNA sequence variation) as the most relevant factor, followed by culture medium (*Figure 1b*). Critically, however, significant donor effects remained after accounting for RNA variability (*Figure 1b,d*; *Figure 1—figure supplement 4*; Materials and methods). This indicates that (i) genetic differences between individual donors and experimental differences between culture conditions play an important role in causing the observed

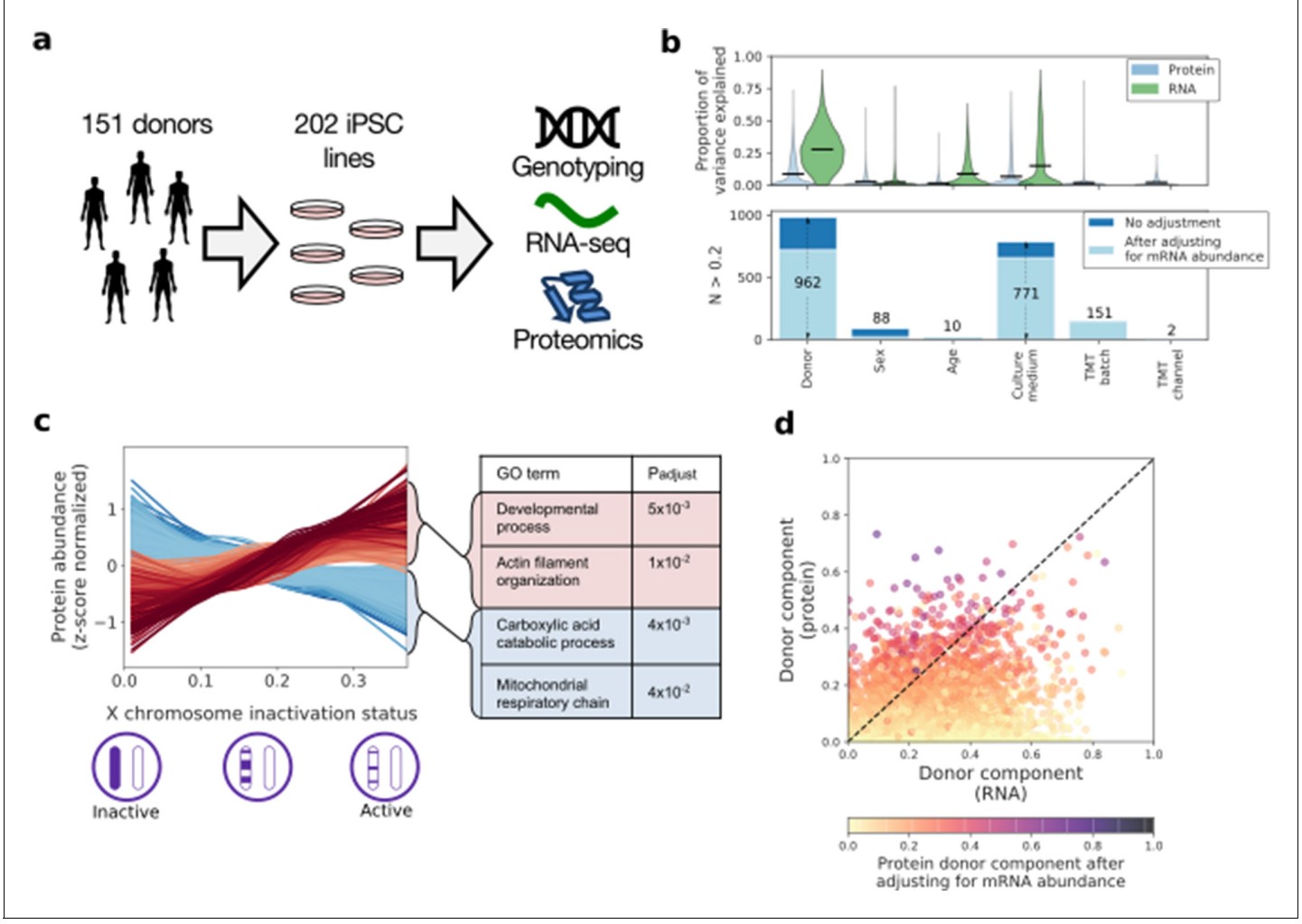

**Figure 1.** | Characterising variation in the iPSC proteome and transcriptome. (a) Experimental design and assays considered in this study. Genotype, RNA-Seq and quantitative proteomics data were obtained from the same cell material of 202 iPSC lines derived from 151 unrelated donors. (b) Variance component analysis of RNA and protein abundances across genes, considering different technical and biological factors. Shown is the distribution of the fraction of variance explained by different factors (upper panel) across proteins, and the number of genes with substantial variance contribution for each factor (>20% contribution; lower panel). Also shown are the number of genes that retain greater than 20% contribution after adjusting for the effect of the corresponding RNA profiles on protein abundance (light blue; see Materials and methods). (c) Association of protein level with X chromosome inactivation (XCI) status across 110 female iPSC lines. Shown are lowess regression curves for 322 and 312 proteins respectively that were identified as significantly up (red) - and down (blue) - regulated with loss of XCI in female iPSC lines (lower panel; 10% FDR). Selected gene ontology enrichments for these sets of proteins are shown (right-hand panel; Materials and methods). XCI status was estimated as the average fraction of allele-specific expression for the inactive chromosome across all chromosome X genes (Materials and methods). (d) Scatter plot of the fraction of variance explained by donor at the RNA (x-axis) versus protein (y-axis) level. Encoded in colour is the fraction of variance explained by donor effects at the protein level after adjusting for the effect of the corresponding RNA profiles on protein abundance (Materials and methods).

The online version of this article includes the following source data and figure supplement(s) for figure 1:

**Source data 1.** HipSci proteomics iPSC lines.
**Source data 2.** RNA gene level expression across the 202 lines for genes recurrently detected at the protein level.
**Source data 3.** Protein abundance values across the 202 and reference lines for genes recurrently detected at the protein level and with RNA expression (TPM >1).
**Source data 4.** Protein and RNA variance components.
**Source data 5.** Protein and RNA correlation with X chromosome inactivation.
**Source data 6.** Functional enrichment of genes with protein or RNA profiles correlated with XCI.
**Figure supplement 1.** Protein quantification, quality control and batch correction.
**Figure supplement 2.** Comparison of iPSC proteome and somatic human tissues.
**Figure supplement 3.** Comparison of the iPS transcriptome and proteome of disease and normal lines.
**Figure supplement 4.** Comparisons of variance component estimates before and after regressing out mRNA effects.

*Figure 1 continued on next page*

*Figure 1 continued*

**Figure supplement 5.** Donor variance components of proteins differentially expressed between iPSC and ESC.
**Figure supplement 6.** Quantification of X chromosome inactivation status in female iPSC lines using chromosome X ASE SNPs.

variation in proteome expression between the iPSC lines and (ii) post-transcriptional mechanisms also contribute to these effects. Notably, many of the proteins showing strong donor effects were previously identified as differentially expressed between reprogrammed iPS cells and embryonic stem cells (ESCs) (*Phanstiel et al., 2011*; *Munoz et al., 2011*), suggesting that some of these previously reported differences could be due to genetic variation between donors, rather than intrinsic differences between the iPSC and ESC cell types (*Figure 1—figure supplement 5*).

The sex of the donor affected proteome expression, including a subset of proteins uniquely encoded on the male-specific X chromosome. There was also a strong (i.e. >20%) gender-related effect on the expression of a subset of 88 proteins (*Figure 1b*), which are enriched for proteins encoded on the X chromosome (Odds ratio = 24.8, PV = $3 \times 10^{-32}$, Fisher's exact test). This reflected the partial erosion of X chromosome inactivation (XCI) observed in a subset of iPSC lines derived from female donors, as confirmed both by quantification of allele-specific expression and *XIST* expression in these lines (*Figure 1—figure supplement 6*). Incomplete XCI has been linked previously to poor iPSC differentiation and changes in RNA levels (*Mekhoubad et al., 2012*; *Salomonis et al., 2016*). However, our data provide the first opportunity to link the XCI status of 110 distinct female iPSC lines (as inferred from allele-specific expression; Materials and methods; *Figure 1—figure supplement 6*), with changes in the abundance of both proteins and RNAs. This identified 1374 genes for which either protein, or RNA levels, or both, showed changes correlated with XCI status (*Figure 1c*, FDR < 10%; *Figure 1—source data 5*). Further analysis indicated that XCI status preferentially impacts catabolic processes and mitochondria at the protein level, while this was not observed at the RNA level (Gene Ontology; Materials and methods; *Figure 1c*; *Figure 1—source data 6*). These data thus reveal an important effect of XCI status on global gene expression in iPSC lines from female donors, including specific effects at the protein level that are not detected by transcriptomic analysis.

## Mapping *cis* genetic effects on protein abundance

Next, we mapped *cis* quantitative trait loci at both the RNA and protein levels, considering 8543 autosomal protein coding genes that were quantified at both levels (MAF >5%; within +/- 250 kb of the gene boundaries; using a linear mixed model; Materials and methods). The number of pQTLs identified was greatly increased by adapting the PEER-based adjustment, which was previously developed for mapping of eQTLs (*Stegle et al., 2012*), for use with proteomic data (Materials and methods; *Figure 2—figure supplement 1*). Across all autosomal genes, we report 654 genes with at least one *cis* pQTL and 3487 genes with a *cis* eQTL (FDR < 10% for both eQTL and pQTL mapping; lead variants only; *Figure 2a*; *Figure 2—source datas 1* and *2*). Among these, 273 genes were shared and had identical or correlated lead variants, whereas 82 genes showed evidence for an eQTL and pQTL with independent lead variants (LD-based criterion, $r^2$ <0.1, *Figure 2—source data 1*). Genes with substantial donor components, as identified based on the variance component analysis (>20%; *Figure 1b*), were enriched for significant *cis* pQTL (215 genes out of 962; Odds ratio = 4.2; *Figure 2—figure supplement 2*).

To identify DNA sequence variants with effects at both the RNA and protein levels, we considered the pairwise replication of pQTLs at the RNA level and vice versa (lead QTL variants, defining 'replication' as nominal PV <0.01 with consistent effect direction; Materials and methods), which is more sensitive than assessing overlapping QTL variants. This identified 473 pQTLs (72%) with replicated eQTL effects. Conversely, 893 eQTLs (26%) had replicated protein effects, with globally concordant effect size directions and distance from gene boundaries (*Figure 2—figure supplement 3*).

Lack of replication of eQTLs at the protein level could arise from a combination of technical and/ or biological factors. We identified the eQTL effect size as a strong predictor for protein replication, with larger effects being associated with increased replication rates (*Figure 2c*). To systematically characterise the determinants of eQTL replication, we considered a random forest model trained to predict the protein replication status (*Figure 2d*). In addition to the eQTL effect size, this identified

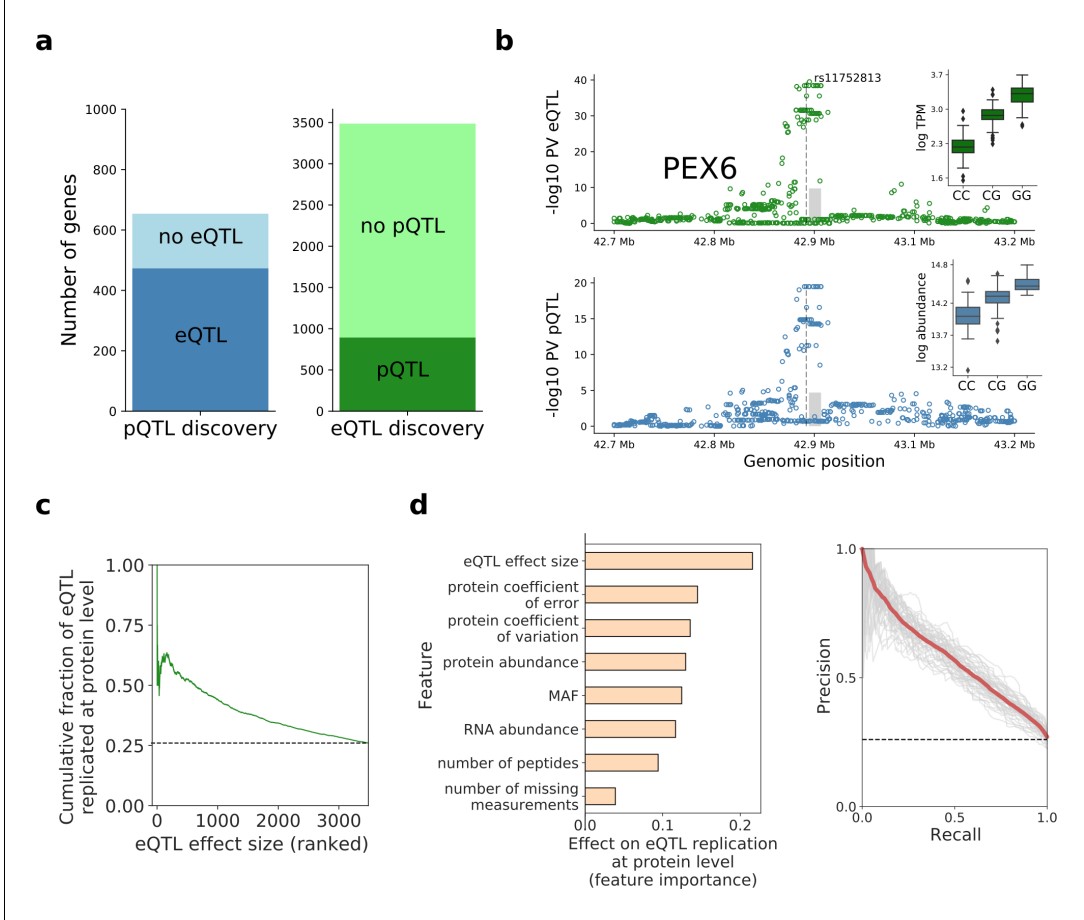

**Figure 2.** Human iPSC *cis* protein and RNA QTLs. (**a**) Number of genes with a protein (blue) or RNA (green) *cis* QTL (FDR < 10%) and pairwise replication of genetic effects. Left: Number of genes with a pQTL, either with (dark blue) or without (light blue) replicated RNA effect. Right: Number of genes with an eQTL, either with (dark green) or without (light green) replicated protein effect. Replication defined by assessing nominal significance (PV <0.01) of QTL in the respective other layer. (**b**) Local Manhattan plots displaying negative log p-values (PV) from *cis* RNA (top) and protein (bottom) QTL mapping for *PEX6*. The dashed line and the grey box indicate the genomic positions of the lead QTL and of the gene. Boxplots show RNA and protein expression for different alleles at the pQTL lead variant rs11752813, a variant in LD ($r^2$ = 1, 1000 Genomes European populations phase 3) with the Alzheimer risk variant rs1129187 (***Jun et al., 2016***) (OR 1.13). (**c**) Cumulative fraction of eQTLs with replicated protein effects as a function of the eQTL effect size (from highest to lowest). (**d**) Prediction of protein replication of eQTLs, considering features derived from gene annotations, eQTL, RNA and protein data. Predictions were obtained using a random forest model trained on the protein replication status of eQTL (as in a; Materials and methods). Left: Feature importance scores. Right: Precision-recall curve for the model, evaluated in independent test fractions. The model performance was assessed by random sampling of training/testing data with a 80/20 split, performed 50 times. Shown in red is the average precision-recall across all sampled training/test splits and in thin grey lines results of individual folds.

The online version of this article includes the following source data and figure supplement(s) for figure 2:

**Source data 1.** pQTL_results.
**Source data 2.** eQTL_results.
**Figure supplement 1.** Selection of the number of PEER factors to adjust for unwanted variation.
**Figure supplement 2.** Relationship between estimates of donor variance component and *cis* pQTLs.
**Figure supplement 3.** Comparison of eQTL and pQTL effect sizes and genomic positions.
**Figure supplement 4.** Example iPS eQTL and pQTL variance with evidence for co-localisation with GWAS variants.

other predictive factors, including the protein coefficient of error (estimated from technical replicate samples; Materials and methods) and the protein coefficient of variation across lines (***Figure 2d***; ***Figure 2—source data 2***).

To explore the physiological relevance of iPSC pQTL variants, we examined their overlap with variants identified in genome-wide association studies (GWAS). Specifically, we probed for QTLs that tag GWAS variants contained in the GWAS catalogue (***MacArthur et al., 2017***) (i.e. are in LD

$r^2$ >0.8), identifying 136 (of 654) pQTL signals that tag a known GWAS variant (*Figure 2—source data 1*). In addition, we assessed the statistical evidence for co-localisation of pQTL and GWAS signals for 51 studies for which full summary statistics were obtained (using eCAVIAR *Hormozdiari et al., 2016*; Materials and methods; *Figure 2—source data 1*), yielding 49 pQTLs with evidence of co-localisation (i.e cumulative co-localisation probability greater than 0.1). Among these, examples of pQTLs with corresponding effects at the RNA level include the variant rs7872034, a pQTL for *SMC2* with co-localisation evidence for serous invasive ovarian cancer (*Phelan et al., 2017*), and the variant rs11752813, a pQTL for *PEX6* and in LD with Alzheimer's disease in APOE e4+ carriers risk variant rs1129187 (*Jun et al., 2016*; *Figure 2—figure supplement 4*; *Figure 2b*).

Notably, for 33 pQTLs linked to GWAS variants, either via co-localisation or LD tagging, no replicated effect was identified at the RNA level, suggesting protein-specific regulation (*Figure 2—source data 1*). For example, rs11601507 has no RNA effects, and is associated with TRIM5 protein abundance and with coronary artery disease risk (*van der Harst and Verweij, 2018*; *Figure 2—figure supplement 4*). Such cases raise the question of the mechanisms by which these variants modulate protein abundance and, ultimately, phenotypic traits, as addressed below.

## pQTL linked to isoform-specific transcript expression

To investigate the mechanisms that underlie discordant eQTLs and pQTLs in more detail, we performed transcript isoform and protein peptide QTL analyses. *cis* QTL mapping of 33,050 reference transcript isoforms (*Zerbino et al., 2018*) (quantified using Salmon *Patro et al., 2017*; Materials and methods) and 119,747 peptides identified 3810 genes with a transcript QTL (tQTL) and 566 genes with a peptide QTL (pepQTL), respectively (*Figure 3—figure supplement 1*, Materials and methods, *Figure 3—source datas 1* and *2*).

Transcript-level QTL mapping could explain the lack of protein effects for a small fraction of the 2594 eQTL without a replicated pQTL effect (*Figure 2a*). For 48 of these, the eQTL variant was identified as exclusively associated with abundance changes of non-coding transcript isoforms (nominal PV <0.01), which explains the absence of protein effects (*Figure 3—figure supplement 2*). Furthermore, when considering 1262 transcript QTL that neither replicate at the eQTL, nor at the pQTL level, in 45 instances we observed consistent replication when considering peptide QTL (*Figure 3—figure supplement 2b*).

Among 181 pQTLs without eQTL replication (*Figure 2a*), 61 had nominally significant transcript QTLs (PV <0.01; *Figure 3a*). For 12 of these, including a pQTL for *MMAB* (*Figure 3b*), we observed genetic effects with opposite directions on coding and non-coding transcript isoforms, which explains the lack of genetic effects when considering gene-level RNA abundance.

## pQTL arising from protein-altering variants

Next, we set out to characterise further the remaining 120 pQTL without replication at either eQTL, or transcript QTL levels. When classifying the corresponding lead pQTL variants based on their predicted functional effect, we identified 24 inframe variants, a striking enrichment compared to pQTL with replicated RNA effects (3.8-fold enrichment; PV = $4 \times 10^{-5}$, Fisher's exact test; *Figure 3c and d*). These findings are in line with previous observations in lymphoblast cell lines (*Battle et al., 2015*). Of note, peptides containing protein-altering variants were excluded from the quantifications (Materials and methods), and the reported pQTL effects were observed for multiple peptides from the same proteins (*Figure 3—figure supplement 3*), providing further confidence in genuine regulatory effects. We assessed whether the 24 pQTL have effects at the RNA level (eQTL) in other cell types, and for 11 of these pQTL we did not find evidence of eQTL nominal significance in any of the 48 GTEx (PV <0.01/48; *Battle et al., 2017*; *Figure 1—source data 1*), which further points to RNA-independent mechanisms.

Inframe variants have the potential to affect protein function. We estimated whether a variant is likely to be deleterious to protein function using SIFT scores, which capture evolutionary conservation and amino acid similarity (*Ng and Henikoff, 2003*). This revealed a clear enrichment of the 24 RNA-independent pQTL that tag inframe variants, 10 of which have predicted deleterious effects (SIFT score <0.05), compared to four among all other pQTL (Odds ratio = 27.5, PV = $3.8 \times 10^{-8}$, Fisher's exact test; *Figure 3d*; *Figure 1—source data 1*). Putative effects of these variants on protein

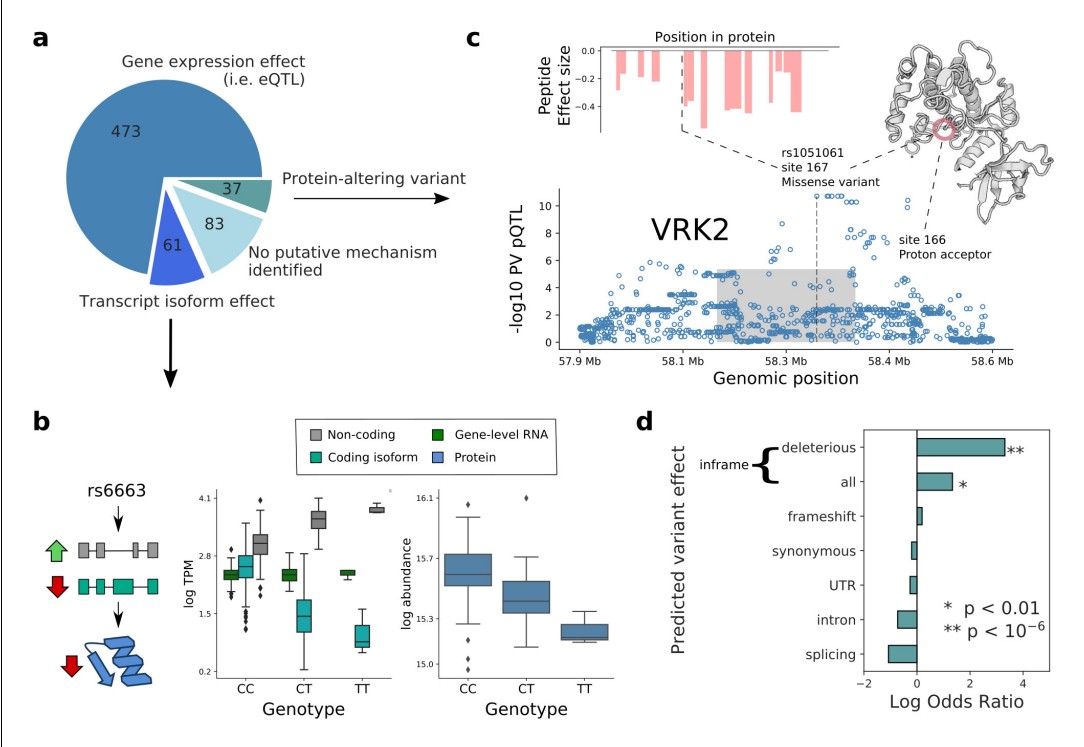

**Figure 3.** Putative mechanisms of pQTL from transcript isoform regulation and protein-altering variants. (a) Categorisation of 654 pQTL into four classes according to their putative mechanism: gene expression effect (i.e. replicated at eQTL level), transcript-isoform specific effect (i.e. not replicated at eQTL level, but significant at transcript isoform level), protein-altering variant (i.e. at least one inframe variant in LD with lead pQTL variant) without expression effect at RNA level, and without any putative mechanism identified. (b) Example pQTL without eQTL replication (rs6663; gene *MMAB*), with a directional opposite effect on a coding and non-coding isoform (cyan: ENST00000540016; grey: ENST00000537496), resulting in no overall change in gene expression level. (c) The pQTL variant (rs1051061) is a protein-altering variant associated with VRK2 protein abundance (below), and lacks detectable effect on RNA expression. The pQTL signal is observed across 15 peptides spanning the VRK2 protein sequence (above, left). This variant is associated with schizophrenia risk, and is located at the kinase active site, proximal to the proton acceptor residue (above, right). The dashed line and the grey box indicate the genomic positions of the lead QTL and of the gene. (d) Enrichment of RNA-independent pQTL in different categories of predicted variant effects, using gene variants in high LD with pQTLs (proxy gene variants; $r^2 > 0.8$; within the *cis* gene boundaries). Enrichment calculated using Fisher's exact test.

The online version of this article includes the following source data and figure supplement(s) for figure 3:

**Source data 1.** tQTL_results.
**Source data 2.** pepQTL results.
**Figure supplement 1.** Discovery and replication of cis QTLs at protein and RNA levels.
**Figure supplement 2.** Isoform-specific genetic effects.
**Figure supplement 3.** Peptide resolution assessment of pQTLs.
**Figure supplement 4.** Quantification of peptides containing coding polymorphisms.

function include loss of enzymatic activity and disruption of protein structure. For example, the variant rs1051061 in VRK2 lies in a conserved sequence in the kinase domain, proximal to the proton acceptor residue, likely impacting kinase activity (*Figure 3c*). The identical variant has been identified as GWAS risk variant for schizophrenia (*Yu et al., 2017*) (OR 1.17), with the risk allele being associated with decreased protein abundance. The effect direction is consistent with previous studies that have linked decreased VRK2 expression to neurological disorders including schizophrenia (*Azimi et al., 2018*; *Tesli et al., 2016*).

These data show important roles of transcriptional regulation underlying *cis* pQTL effects, while also highlighting how isoform-specific effects, which are invisible to standard eQTL mapping approaches, can be detected at the protein level. For a substantial subset of pQTLs, we identified linked protein-altering variants, many with deleterious effects. Together with previous observations,

these results suggest that proteomics information can aid understanding of pathogenic mechanisms of deleterious variants.

## Proteome-wide effects of cis QTLs

Building on the compendium of *cis* pQTL identified here in iPS cells, we set out to characterise downstream proteome-wide changes. We mapped proteome-wide *trans* pQTL, considering 654 *cis* pQTL variants. This identified 51 *cis*-pQTL lead variants with *trans* effects on a total of 68 proteins (FDR < 10%; *Figure 4—source data 1*; Materials and methods). To rule out synthetic associations, we discarded associations with evidence for sequence similarity between *cis* and *trans* proteins, and we verified the consistency of the identified *trans* effects across multiple independent peptides (Materials and methods; *Figure 3—figure supplement 3*). The detected pairs of proteins with shared genetic regulation were strongly enriched for known protein-protein interactions (CORUM *Ruepp et al., 2010*, IntAct *Orchard et al., 2014*, StringDB *Szklarczyk et al., 2017*; Odds ratio = 9.1, PV = $1.5 \times 10^{-10}$, Fisher's exact test; *Figure 4b*). The *cis* and *trans* effects had similar effect directions and effect sizes, consistent with genetic effects mediated via stabilising protein-protein interactions (*Figure 4c*). This interpretation of our data in human iPSCs is consistent with the significant donor variance component we observed for many protein complexes (*Figure 4d*). It is also consistent with previous observations in an outbred mouse cross, showing that protein modules sharing genetic effects in *trans* are enriched in protein interactions (*Ruepp et al., 2010*), and identification of *trans* protein effects due to somatic aberrations in human cancer cell lines (*Gonçalves et al., 2017*; *Roumeliotis et al., 2017*). Importantly, our results generalise these previous observations to genetic effects of common variants that segregate in human populations.

In summary, the *trans* effects we detected appear to induce strong correlations across protein complex subunits (*Figure 4d*), whereby a variant associated in *cis* with one subunit was also associated in *trans* with other subunits. This is illustrated by PEX26-PEX6-PEX1, a protein complex involved in peroxisome biogenesis. As noted above, the underlying pQTL variant *rs1129187* is associated in *cis* with an increase in both PEX6 RNA and protein abundance (*Figure 2b*) and is a known risk variant for Alzheimer's disease in APOE e4+ carriers (*Jun et al., 2016*). This *cis pQTL* in turn induces downstream associations on the remaining complex subunits, PEX26 and PEX1 (*Figure 4e*), suggesting that PEX6 acts as a limiting subunit of this complex in iPSCs. Thus, our results provide a potential biological mechanism underlying this risk variant, namely acting through changes in the abundance of the PEX26-PEX6-PEX1 complex. Notably, there is prior evidence for an implication of peroxisomal function in the development of Alzheimer's disease and in other neurodegenerative processes (*Lizard et al., 2012*; *Berger et al., 2016*), providing further support for this hypothesis.

## Discussion

Here, we report the first in-depth characterisation of the human iPSC proteome, connecting genetic variation to changes in RNA and protein levels. Beyond the relevance for iPS cell biology, this study, to our knowledge, provides the most detailed population-level analysis of parallel RNA/protein profiles in human cells. By quantifying genome-wide protein and transcript expression variation across more than 200 human iPSC lines, we identified both genetic and non-genetic mechanisms that underlie variation in both protein and RNA levels. We have mapped more than 600 *cis* protein quantitative trait loci (pQTLs) and analysed how these relate to *cis* eQTLs, how they impact other proteins in *trans*, and how pQTLs link to human disease variants.

The variance component analysis explained a lower overall fraction of variance in the protein data compared to RNA variation, which likely reflects larger technical effects and/or stochasticity in protein expression levels. Among the explainable fraction of variance, donor-specific genetic factors are a major contributor to the differences in protein expression detected across the iPSC lines. The corollary is that protein expression variation across iPS cells reflects genetic diversity in the human population. Consistent with this, we identified 654 common genetic variants associated with changes in protein abundance.

Globally, there were substantially fewer pQTLs than eQTLs, and while most pQTLs had effects of the same direction at eQTL, only 30% of eQTLs are nominally significant at the protein level. It is possible that technical factors resulting from the protein measurement methods may contribute, at least in part, towards attenuating the signal detected at the protein level. However, considering our

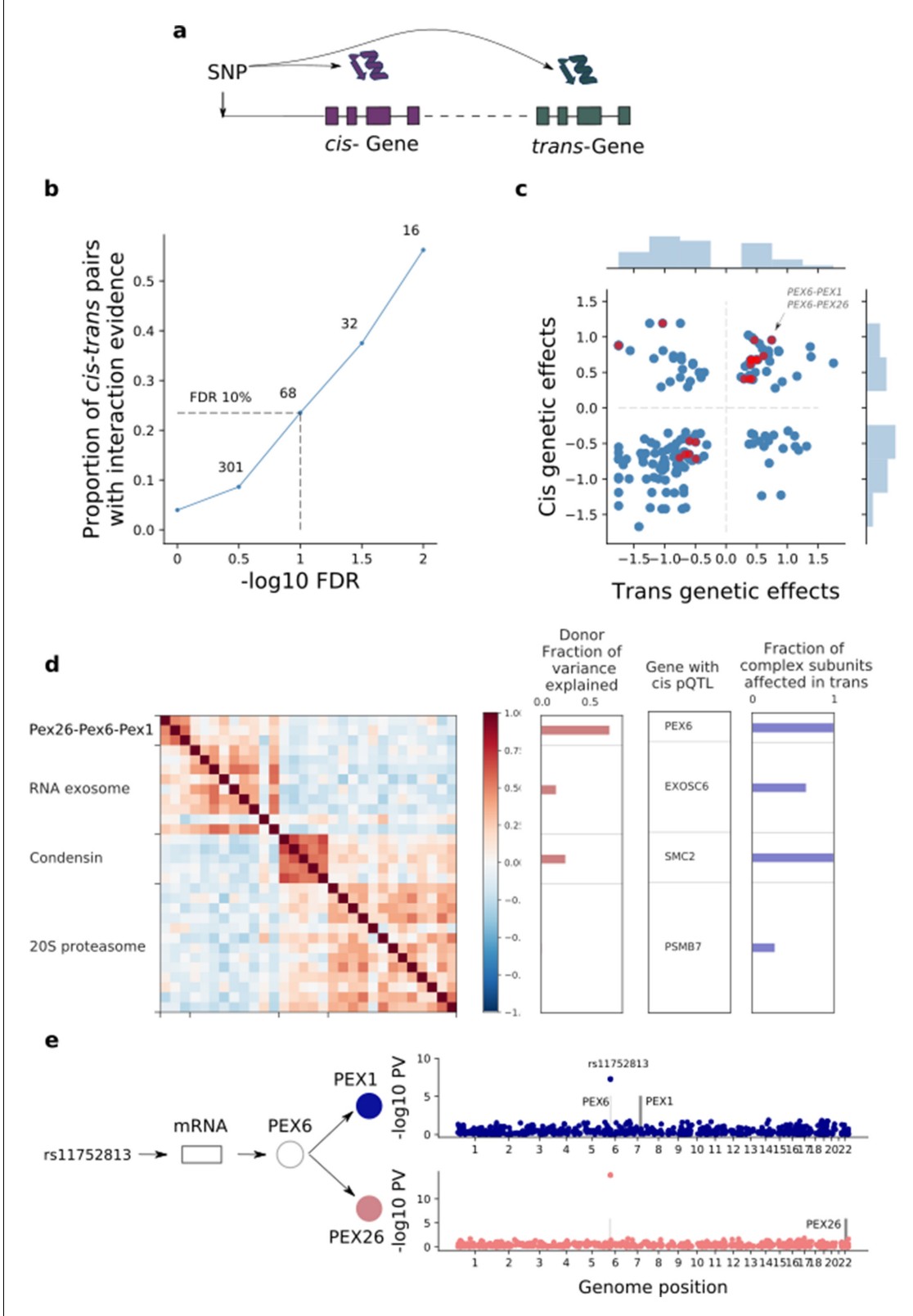

**Figure 4.** | *Trans* effects on the iPSC proteome. (a) Strategy for mapping *trans* genetic effects on protein abundance. Lead *cis* pQTL variants were considered for proteome-wide association analysis. (b) Enrichment of previously catalogued protein-protein interactions among significant *trans* pQTLs. Shown is the fraction of *cis-trans* gene pairs linked by a *trans* pQTL with evidence of protein-protein interactions (based on the union of CORUM, IntAct, and StringDB), as a function of the considered FDR threshold for *trans* pQTL discovery. The dashed lines correspond to FDR < 10%. Numbers

*Figure 4 continued on next page*

*Figure 4 continued*

indicate the number of *trans* pQTL identified for each FDR threshold. (c) Comparison of genetic effect sizes, in cis and *trans*, for significant (FDR < 10%) *trans* pQTLs. Red points indicate *cis-trans* pairs with evidence for protein-protein interactions defined as in b. (d) Left: Protein co-expression of protein complex subunits defined based on CORUM. Right: i) subunit with the most significant *cis* pQTL; ii) fraction of subunits in association with the *cis* pQTL at nominal significance (PV <0.01). iii) fraction of the average cluster protein expression level explained by donor effects. (e) *Trans* regulation of the PEX26-PEX6-PEX1 complex. The variant *rs11752813* (LD r$^2$ = 1 with *rs1129187*) is associated in cis with changes in the RNA and protein abundance of PEX6 and in trans with changes in the protein abundance of PEX1 and PEX26.

The online version of this article includes the following source data for figure 4:

**Source data 1.** trans-pQTL_results.

data in light also of results from previous studies, some of which employed alternative technologies for protein detection to the MS methods used here, we suggest that the signal attenuation between eQTL and pQTL levels is not exclusively the result of limitations in protein measurements. Instead, many eQTLs may reflect variation in RNA abundance that does not cause significant changes in steady state protein levels.

By the systematic comparison of matched protein and RNA data, including detailed analysis of separate isoforms, we demonstrated that in order to fully understand the propagation of genetic effects to proteins, isoform-resolution protein and RNA phenotypes are indispensable. In particular, this approach identified additional RNA-dependent regulation that manifests in protein QTL, thereby improving the ability to identify genuine RNA-independent pQTL.

We showed that the pQTLs for which no corresponding changes in transcript levels were detected, are enriched in deleterious missense variants. This result suggests that the phenotypic effects of such variations may be exerted through protein abundance changes. Because most deleterious variants, and in particular pathogenic variants, are rare, larger sample sizes will be required to fully assess the protein components of this class of regulatory genetic effects.

Our study presents the first comprehensive map of pQTLs at peptide resolution, considering a total of 119,747 peptides from 8543 proteins for genetic analysis. This identified 566 peptide QTL, several of which were not detectable when considering whole protein expression levels, as illustrated with the variant rs12795503, pepQTL for gene CTTN (*Figure 3—figure supplement 1*). While we mapped fewer significant pepQTLs than pQTLs, peptide level analyses were shown here to overcome potential artefacts raised by protein quantification, in particular when mapping *trans* pQTL, and are invaluable in identifying isoform-specific effects.

Our data highlight the ability of protein-protein interactions to propagate genetic effects in human populations. A long-standing hypothesis has been that certain protein complexes may have a rate-limiting subunit that determines complex abundance, with any excess subunits produced being rapidly degraded (e.g. because of exposure of hydrophobic residues). This implies that *cis* eQTLs affecting the levels of rate-limiting subunits should also have effects in *trans* on the abundance of the whole complex, and on most, if not all, subunits therein. While *trans* genetic effects were previously reported to be mediated by protein interactions in high heterozygosity samples, that is outbred mice, (*Chick et al., 2016*) and for somatic aberrations in cancer cell lines (*Gonçalves et al., 2017*; *Roumeliotis et al., 2017*), to our knowledge, this study provides the first example that such effects act through common genetic variants in untransformed human cells. In the future, the approach we have taken here could be extended by mendelian randomisation-based approaches to formally assess the mediating role of the *cis* pQTL on protein complex members.

Understanding the mechanisms through which genetic variations act in the human population is of great relevance to characterising risk factors and susceptibility to disease. There is on-going interest in the potential for studying disease mechanisms using disease relevant tissues that are derived from panels of iPSCs (*Cayo et al., 2017*; *Li et al., 2018*; *D'Aiuto et al., 2014*; *Schwartzentruber et al., 2018*). This study provides important information showing how direct analysis of human iPSCs can advance our understanding of the genetic regulation of protein expression and how this influences cell phenotypes and disease mechanisms.

# Materials and methods

### Key resources table

| Reagent type (species) or resource | Designation | Source or reference | Identifiers | Additional information |
|---|---|---|---|---|
| Cell line (*Homo-sapiens*) | iPSC | www.hipsci.org | RRID:SCR_003909 | |
| Software, algorithm | MaxQuant | https://www.maxquant.org/ | RRID:SCR_014485 | |
| Software, algorithm | Trim Galore | https://www.bioinformatics.babraham.ac.uk/projects/trim_galore/ | RRID:SCR_011847 | |
| Software, algorithm | STAR | https://www.ncbi.nlm.nih.gov/pmc/articles/PMC3530905/ | RRID:SCR_015899 | |
| Software, algorithm | Salmon | https://combine-lab.github.io/salmon/ | NA | |
| Software, algorithm | g:Profiler | http://biit.cs.ut.ee/gprofiler/ | RRID:SCR_006809 | |
| Software, algorithm | eCAVIAR | http://zarlab.cs.ucla.edu/tag/ecaviar/ | NA | |
| Software, algorithm | VEP | https://www.ensembl.org/info/docs/tools/vep/index.html | RRID:SCR_007931 | |
| Software, algorithm | MutFunc | http://www.mutfunc.com/ | NA | |
| Software, algorithm | Limix | https://github.com/limix/limix | NA | |
| Software, algorithm | Peer | http://www.sanger.ac.uk/resources/software/peer/ | RRID:SCR_009326 | |
| Software, algorithm | Scikit-learn | http://scikit-learn.org/ | RRID:SCR_002577 | |

## Cell lines

As described in *Kilpinen et al., 2017*, all HipSci samples were collected from consented research volunteers recruited from the NIHR Cambridge BioResource, and iPSC were generated from fibroblasts by transduction with Sendai vectors. In brief, cells were cultured on fibroblasts (MEF-CF1) feeder layer with selected lines being transferred on Essential 8 (E8) medium. Pluripotency was assessed based on expression profiling, detection of pluripotency markers in culture and response to differentiation inducing conditions. Mycoplasma screening was performed using a standard PCR kit. Sample identity was confirmed using a Fluidigm genotyping assay containing 24 SNPs. The ID numbers and details for each cell line included in this analysis are listed in *Figure 1—source data 1*.

## RNA-Seq data processing

Raw RNA-Seq data for 331 samples were obtained from the ENA project: ERP007111. CRAM files were merged on a sample level and were converted to FASTQ format. Sequencing reads were trimmed to remove adapters and low-quality bases (using Trim Galore! *Dobin et al., 2013*), followed by read alignment using STAR (version: 020201) (*Liao et al., 2014*), using the two-pass alignment mode and the default parameters as proposed by ENCODE (c.f. STAR manual). All alignments were relative to the GRCh37 reference genome, using ENSEMBL 75 as transcript annotation (*Zerbino et al., 2018*).

Samples with low-quality RNA-Seq were discarded, if they had less than 2 billion bases aligned, had less than 30% coding bases, or had a duplication rate higher than 75%. This resulted in 323 lines for analysis, of which 202 had matched proteome data.

Gene-level RNA expression was quantified from the STAR alignments using featureCounts (v1.6.0) (*Robinson et al., 2010*), which was applied to the primary alignments using the '-B' and '-C' options in stranded mode, using the ENSEMBL 75 GTF file. Quantifications per sample were merged into an expression table using the following normalisation steps. First, gene counts were normalized by gene length. Second, the counts for each sample were normalised by sequencing depth using the edgeR adjustment (*McAlister et al., 2014*).

Transcript isoform expression was quantified directly from the (unaligned) trimmed reads using Salmon (*Zerbino et al., 2018*) (version: 0.8.2), using the '–seqBias', '–gcBias' and 'VBOpt' options in 'ISR' mode to match our inward stranded sequencing reads. The transcript database was built on transcripts derived from ENSEMBL 75. The TPM values as returned by Salmon were combined into an expression table.

## Quantitative proteomics data generation

Upon establishment in culture, lines were expanded for banking and molecular assays (including genomics, transcriptomics and proteomics). We selected 217 lines (banked frozen pellets) for in-depth proteomic analysis with Tandem Mass Tag Mass Spectrometry. A subset of 202 lines (112 normal and 90 disease; *Figure 1—source data 1*) with matched mRNA and protein data were considered for further analysis. Previous studies have shown that this sample size enables the identification of a large number of *cis* RNA and protein QTLs (*Battle et al., 2015*).

## Sample preparation

For protein extraction, frozen iPSC cell pellets were washed with ice cold PBS and redissolved immediately in 200 µL of lysis buffer (8 M urea in 100 mM triethyl ammonium bicarbonate (TEAB) and mixed at room temperature for 15 min. DNA in the cell lysates was sheared using ultrasonication (6 × 20 s at 10°C). The proteins were reduced using tris-carboxyethylphosphine TCEP (25 mM) for 30 min at room temperature, then alkylated in the dark for 30 min using iodoacetamide (50 mM). Total protein was quantified using the fluorescence based EZQ assay (Life Technologies). The lysates were diluted four-fold with 100 mM TEAB for the first protease digestion with mass spectrometry grade lysyl endopeptidase, Lys-C (Wako, Japan), then diluted a further 2.5-fold before a second digestion with trypsin. Lys-C and trypsin were used at an enzyme to substrate ratio of 1:50 (w/w). Digestions were carried out for 12 hr at 37°C, then stopped by acidification with trifluoroacetic acid (TFA) to a final concentration of 1% (v:v). Peptides were desalted using C18 Sep-Pak cartridges (Waters) following manufacturer's instructions and dried.

## Tandem Mass Tag mass spectrometry analysis

For Tandem Mass Tag (TMT)-based quantification, the dried peptides were redissolved in 100 mM TEAB (50 µL) and their concentration was measured using a fluorescent assay (CBQCA) (Life Technologies). 100 µg of peptides, from each cell line to be compared, in 100 µL of TEAB were labelled with a different TMT tag (20 µg ml$^{-1}$ in 40 µL acetonitrile) (Thermo Scientific), for 2 hr at room temperature. After incubation, the labelling reaction was quenched using 8 µl of 5% hydroxylamine (Pierce) for 30 min and the different cell lines/tags were mixed and dried in vacuo. TMT-ten plex was used to label ten IPSC lines and quantify them in parallel. In total, 24 TMT-ten plex experiments were performed, where one IPSC line (HPSI0314i-bubh_3) was chosen as a reference cell line and was kept constant in all TMT batches. The other nine quantification channels were used to label 9 different cell lines. No randomisation was applied in assigning the samples to batches.

The TMT samples were fractionated using off-line, high pH reverse phase chromatography: samples were loaded onto a 4.6 × 250 mm Xbridge BEH130 C18 column with 3.5 µm particles (Waters). Using a Dionex bioRS system, the samples were separated using a 25 min multistep gradient of solvents A (10 mM formate at pH 9) and B (10 mM ammonium formate pH 9 in 80% acetonitrile), at a flow rate of 1 ml/min. Peptides were separated into 48 fractions, which were consolidated into 24 fractions. The fractions were subsequently dried and the peptides redissolved in 5% formic acid and analysed by LC-MS.

5% of the material was analysed using an orbitrap fusion tribrid mass spectrometer (Thermo Scientific), equipped with a Dionex ultra high-pressure liquid chromatography system (nano RSLC). RP-LC was performed using a Dionex RSLC nano HPLC (Thermo Scientific). Peptides were injected onto

a 75 μm × 2 cm PepMap-C18 pre-column and resolved on a 75 μm × 50 cm RP- C18 EASY-Spray temperature controlled integrated column-emitter (Thermo), using a 4-hr multistep gradient from 5% B to 35% B with a constant flow of 200 nL min$^{-1}$. The mobile phases were: 2% ACN incorporating 0.1% FA (Solvent A) and 80% ACN incorporating 0.1% FA (Solvent B). The spray was initiated by applying 2.5 kV to the EASY-Spray emitter and the data were acquired under the control of Xcalibur software in a data-dependent mode using top speed and 4 s duration per cycle. The survey scan is acquired in the orbitrap covering the *m/z* range from 400 to 1400 Th, with a mass resolution of 120,000 and an automatic gain control (AGC) target of 2.0 e5 ions. The most intense ions were selected for fragmentation using CID in the ion trap with 30% CID collision energy and an isolation window of 1.6 Th. The AGC target was set to 1.0 e4 with a maximum injection time of 70 ms and a dynamic exclusion of 80 s.

During the MS3 analysis for more accurate TMT quantifications, five fragment ions were co-isolated using synchronous precursor selection, using a window of 2 Th and further fragmented using a HCD collision energy of 55% (*McAlister et al., 2014*). The fragments were then analysed in the orbitrap with a resolution of 60,000. The AGC target was set to 1.0 e5 and the maximum injection time was set to 105 ms.

## Proteomics data processing

The TMT-labelled samples (24 batches of TMT-ten plex) were analysed using MaxQuant v. 1.6.0.13 (*Cox et al., 2011*; *Schwanhäusser et al., 2011*). Proteins and peptides were identified using the Uni-Prot *human* reference proteome database (Swiss Prot + TrEMBL) release-2017_03, using the Andromeda search engine. Run parameters and the raw MaxQuant output have been deposited at PRIDE (PXD010557).

The following search parameters were used: reporter ion quantification, mass deviation of 6 ppm on the precursor and 0.5 Da on the fragment ions; Tryp/P for enzyme specificity; up to two missed cleavages, 'match between runs', 'iBAQ'. Carbamidomethylation on cysteine was set as a fixed modification. Oxidation on methionine, pyro-glu conversion of N-terminal Gln, deamidation of asparagine and glutamine and acetylation at the protein N-terminus, were set as variable modifications (*Cox and Mann, 2008*; *Cox et al., 2011*; *Schwanhäusser et al., 2011*; *Tyanova et al., 2016*).

Peptides and protein groups, that is groups of protein isoforms identified by common peptides (for details see *Brenes et al., 2018*), were reported at FDR < 5%. The same FDR threshold was used for reporting the Peptide Spectrum Matches (PSM). We performed the FDR calculation on an extended set and removed the Razor Protein FDR calculation constraint (for more details see reference *Ramasamy et al., 2013*). In total, we identified 255,015 peptides detected in at least one sample (after removing reverse and contaminant peptides; on the 217 lines and 23 replicates of the reference line), which corresponds to 16,773 protein groups.

## Quality control and quantification

Peptides that overlap non-synonymous variants may be incorrectly detected and could result in synthetic associations, similarly to the polymorphism-in-probe effects in microarrays (*Schlaffner et al., 2017*). We performed the following quality control and filtering steps. First, using PoGo; (*Lippert et al., 2015*) (applied with Gencode release 30 mapped on GRCh 37), we reconstructed the genomic loci of 250,171 peptides that could be assigned to one or multiple genomic locations. Next, we assessed the overlap of these peptides with common, non-synonymous variants in human populations (MAF >0.01 in 1000 Genomes European populations phase 3). To define non-synonymous variants, we considered the overlap with any transcript isoform that contains the analysed peptide obtained from Gencode release 30 mapped on GRCh 37. This resulted in 6273 peptides that overlap such polymorphisms, which were discarded from further analyses (*Supplementary file 3*). Peptides mapped to multiple locations in the genome were discarded if they overlap non-synonymous variants for at least one of these locations. Finally, we discarded 4844 peptides for which a genetic position was not identified.

We discarded 10 lines with fewer than 67,000 identified peptides (corresponding to 75% of the median number of peptides identified; *Figure 1—figure supplement 1*), resulting in a proteomics dataset consisting of 207 lines, 202 of which had matched RNA-Seq data and hence were considered

for further analysis. In addition, the technical replicates for the included reference line in each TMT batch were retained to aid the normalisation of protein quantifications between batches; see below.

Protein group abundances were estimated using the remaining peptides as the sum of the intensities of individual peptides mapped to the protein group. Peptide abundance estimates were obtained from the intensity values reported in the 'Peptides' file from MaxQuant.

For downstream analysis, we considered the subset of peptides that were recurrently detected in at least 30 of the 202 lines. Similarly, we discarded all protein groups that did not contain at least one recurrent peptide. This resulted in a final dataset of 11,140 recurrent protein groups and 132,078 recurrent peptides (*Figure 1—figure supplement 1*), corresponding to 10,198 genes (Ensembl ID).

To adjust for technical effects during the acquisition of protein data in TMT batches, we scaled the abundance estimate for each feature (i.e. protein or peptide) as follows. For each feature and batch, we multiply the intensity with a scaling coefficient defined as the ratio between the median intensity across all lines and the median across the subset of lines within a given TMT batch. Next, we employed quantile normalisation for peptide and protein abundance estimates, by performing quantile normalisation of the feature distribution in each line relative to a normalisation reference line (the line with the highest number of total peptides detected). Briefly, for each line we assigned for each feature the value observed in the reference line corresponding to the rank of the value of that feature in the line to be normalised: $y'_{pl} = r\,[rank\ y_{pl}]$, where $y_{pl}$ are the intensity values for feature p and line l obtained after batch scaling, $r$ is the sorted vector of intensities from the normalisation reference line, and $y'_{pl}$ is the normalised value. In order to evaluate how the data transformation tackles batch effects, we performed a PCA analysis of protein quantifications and compared the peptide coefficient of error before and after data transformation (*Figure 1—figure supplement 1d and e*).

## Assessment of TMT ratio compression effects

We assessed quantitative compression of our proteomics data by examining changes in peptide abundance for those peptides that were discarded because they overlap non-synonymous variants, following *Battle et al., 2015*. The rationale behind this approach is that a non-synonymous variant in a peptide prevents detection of that peptide, as its sequence will not exist in the proteome reference. Thus, in samples heterozygous for the non-synonymous variants, the measured peptide abundance is expected to be half of that of samples homozygous for the reference variant. Although the data indicated that ratio compression effects can be noticed in our study (*Figure 3—figure supplement 4*), the QTL results show that protein measurements derived from TMT Mass spectrometry analyses are suitable for the detection of protein abundance changes.

## Comparisons of iPSC proteome profiles to existing datasets

To compare our iPSC proteome dataset to the Human Proteome Map (HPM) (*Kim et al., 2014*; *Figure 1—figure supplement 1d*), we first mapped the RefSeq IDs of proteins quantified in the HPM to UniProt IDs. We considered the subset of 8333 proteins that were expressed in our iPSC dataset and in at least one HPM tissue, and for which IDs could be mapped. We then calculated spearman correlation coefficients between the aggregate iPSC proteome abundance profile (averaged across lines) and each HPM tissue.

## Variance component analysis

In order to calculate the contribution of each factor k to variation in protein abundance, we fitted a random effects model as follows: $y = \mu + \sum_k u_k + \varepsilon$ ; $u_k \sim N(0, \sigma_k^2 \cdot M_k)$; $\varepsilon \sim N(0, \sigma_r^2 \cdot I)$; $M_k\,[i, j] = \{1$ if $f_k\,[i] = f_k\,[j]$; $0$ if $f_k\,[i] \neq f_k\,[j]\}$.

where y denotes the (N x 1) vector of normalised log protein abundances, $u_k$ are the random effects, $M_k$ is the (N x N) covariance structure, $\sigma_k$ is the standard deviation, and $\varepsilon$ is the residual (i.i.d. noise). The random effect components are defined based on a categorical covariance function defined on covariates $f_k$, that is the vector of observed values for factor k (e.g. $f_k\,[i] \in \{'male', 'female'\}$ when k is the donor sex component). We considered donor identity, donor sex, donor age (treated as a categorical variable, with each group in age windows of 5 years), culture medium, TMT batch, and TMT channel as random effect components. In order to accurately estimate the donor

variance component, we restricted this analysis to the set of lines from the subset of 51 donors for which two cell lines were assayed and on 6009 genes with TPM >1 and with proteins detected in this subset of lines. Analogous analyses were considered for RNA abundance, leaving out the TMT-specific, random effects.

In order to study the effect of different factors on protein abundance, after adjusting for the effects of RNA abundance on protein abundance, we also applied the variance decomposition analysis to protein abundance values after adjusting for RNA variation. Adjusted protein abundances were calculated by regressing out the effects of RNA abundance (i.e. gene-level quantifications of RNA), on protein abundance for each RNA-protein pair. To do this, we fitted a linear model between RNA and protein abundances across lines (using the Numpy function poly1d in Python), taking the model residuals as the adjusted protein abundance values. Variance decomposition models were then fitted as described above.

All variance component models were fitted using the LIMIX package (https://github.com/PMBio/limix; https://doi.org/10.1101/003905) (*Reimand, 2016*).

## Quantification of X chromosome inactivation (XCI)

The X chromosome inactivation (XCI) status of female cell lines was quantified using allele-specific counts from RNA-Seq reads mapping to the X chromosome. These allele-specific counts were obtained for SNPs present in DBSNP using GATK ReadCounter with the command 'GenomeAnalysisTK.jar -T ASEReadCounter -U ALLOW_N_CIGAR_READS –minMappingQuality 10 –minBaseQuality 2'. For known heterozygous SNPs in each line, the allele-specific fraction of expression was defined as the fraction of reads mapping to the less expressed allele (i.e. the allele-specific fraction was ≤0.5). The XCI status of each cell line was then defined as the mean of the allele-specific fractions across all heterozygous X chromosome SNPs with at least 20 overlapping reads in the corresponding RNA-Seq sample.

Gene ontology enrichment was performed against the 6335 genes included in the XCI analysis using g:Profiler (*Ongen et al., 2016*), and p-values were adjusted for multiple testing using the Benjamini-Hochberg FDR procedure.

## QTL mapping of RNA and protein traits
### *Cis* QTL mapping

We used PEER (*Stegle et al., 2012*) to account for unwanted variation and confounding factors. PEER was applied to log normalised protein abundance and log normalised gene TPM, considering the most highly expressed 10,000 proteins and genes, respectively. We fit 7 PEER factors for protein and 13 PEER factors for RNA, settings that were determined as the largest number of PEER factors that retain statistical independence of the inferred factors (R < 0.7; *Figure 2—figure supplement 1*).

For *cis* genetic analyses, we considered common variants (MAF >5%) in gene-proximal regions of 250 k upstream and downstream of gene transcription start and end sites (GRCh37). The chosen size of this *cis* analysis window is a compromise between comprehensiveness to detect distal regulatory elements, while managing the multiple testing burden. We used a linear mixed model implemented in LIMIX, thereby controlling for both population structure and repeat lines from the same donor, using kinship as a random effect component. The population structure random effect was accounted for using the realized relationship covariance,that is dot product of the genotype matrices. PEER factors were included as fixed effect covariates.

To adjust for multiple testing across *cis* variants for each gene, we fit an empirical null distribution using a permutation scheme combined with a parametric fit to the null distribution, similar to the approach taken in Fast QTL (*Walter et al., 2015*), Briefly, for each gene, we obtained p-values from 100 permutations of the tested variants. We then estimated an empirical null distribution by fitting a parametric Beta distribution to the obtained p-values. Using this null model, we estimated *cis* region adjusted p-values for QTL lead variants. For multiple testing adjustment across genes, we performed Benjamini-Hochberg adjustment.

For protein, peptide and transcript QTLs, as multiple of these features map to the same gene, we used Bonferroni adjustment to correct for feature multiplicity for each gene, followed by Benjamini-Hochberg adjustment, as performed for the gene-level eQTLs.

*Trans* QTL mapping

For *trans* QTLs analysis, we considered lead *cis* QTLs (FDR < 10%; 654 pQTLs) versus 11,140 recurrently detected proteins. Genome-wide adjustment for multiple testing was performed using Benjamini Hochberg (BH) across all tests ($7 \cdot 10^6$ variants $\times$ proteins).

To rule out any potential artefact linked to the mis-mapping of *cis* protein peptides to the *trans* proteins, we aligned all *trans* peptide sequences to the *cis* protein sequences. We considered all peptides used for the quantification of proteins associated in trans and locally aligned them to the reference sequence of the proteins associated in cis (using pairwise2.align.localxs from the Biopython Project, with a gap penalty of 1). This identified a single *trans* association for which the peptides had less than two mismatches, which was excluded from the reported results.

## Downstream analysis of QTL results

### *Cis* eQTL and pQTL replication

To assess the replication of QTL across molecular layers, we considered the QTL detected in one layer and assessed the nominal significance in the other layer (PV <0.01), as well as the consistency of the effect directions in the second layer.

To identify the determinants of eQTL to pQTL replication, we trained a Random Forest model to the replication status of 3487 eQTL (*Figure 2—source data 2*). For each RNA-protein pair, we defined eight features: 'eQTL effect size', 'protein coefficient of error', 'protein coefficient of variation', 'protein abundance', "SNP MAF", 'RNA abundance', 'number of peptides', and 'number of missing measurements'. The feature 'protein coefficient of error' was computed as the coefficient of variation of the reference line across the set of technical replicates (TMT batches). This model was fit in Python using the scikit-learn library v0.21.3, with the sklearn.ensembl.RandomForestClassifier model with n_estimators = 100 (i.e. the suggested default). The model was trained and tested 50 times, training on a random sample of 80% of the data, and tested on the remaining 20%.

## Annotation of *cis-trans* protein pairs with protein-protein interactions

Protein-protein interactions were obtained from the union of CORUM (*Ruepp et al., 2010*), IntAct (*Orchard et al., 2014*) and protein-binding interactions from StringDB (*Szklarczyk et al., 2017*). For CORUM, we considered pairwise interactions between all protein complex subunits, discarded any isoform extension from the protein UniProt IDs, and intersect *cis-trans* protein pairs with the protein-protein interactions reference list.

## Overlap with disease variants and GTEx eQTLs

Following the approach in *Kilpinen et al., 2017*, we defined proxy variants of each cis QTL as variants in high LD ($r^2$ >0.8) based on the UK10K European reference panel (*Buniello, 2019*) and located in the same *cis* window. A QTL was defined as GWAS-tagging if at least one proxy variant was annotated in the NHGRI-EBI GWAS Catalog (download on 2019–03) (*Staley et al., 2016*). A pQTL was defined as replicating in GTEx eQTLs if it was mapped at nominal significance (PV <0.01/48) in any of the 48 tissues (*Battle et al., 2017*).

Complementary, we considered a more stringent criterion based on statistical co-localisation of GWAS signals with QTLs. Specifically, we used summary statistics from phenotypic traits mapped in 51 studies (*McLaren et al., 2016*), using eCAVIAR (*Hormozdiari et al., 2016*) to test for co-localisation. For each pQTL *cis* region and each GWAS trait and study we first intersected the variants assessed both in GWAS and in our study, resulting between $10^4$ and $2 \times 10^5$ variants. We then selected the traits with at least one variant with $PV_{GWAS}$ <$10^{-5}$ and genome wide-significant in our study. For each trait - QTL pair we performed the co-localisation of z-scores (computed as the ratio between effect size and effect size standard deviation).

## Annotation of pQTLs with variant effects

For each pQTL lead variant, a set of proxy variants were identified based on LD ($r^2$ >0.8). Each proxy variant located in the gene body for the corresponding gene was annotated based on its position and predicted effect, using Variant Effect Predictor (*Eilbeck et al., 2005*). The variants were grouped in parent categories using Sequence Ontology (2016 release; *Wagih et al., 2018*) as follows: 'inframe_variant' includes variants annotated as 'inframe_deletion', 'inframe_insertion',

'incomplete_terminal_codon_variant', 'stop_lost', 'stop_gained' and 'missense_variant'; 'splicing_variant' includes variants annotated as 'splice_acceptor_variant', 'splice_donor_variant' and 'splice_region_variant'; 'frameshift_variant' includes variants annotated as 'feature_elongation', 'feature_truncation' and 'frameshift_variant'. The set of inframe variants were further classified as deleterious or not according to their SIFT scores (*Ng and Henikoff, 2003*), as provided by MutFunc [69] (SIFT score <0.05 corresponds to a deleterious mutation). Each pQTL SNP with at least one proxy variant was annotated with the predicted effects of all its proxy variants. For each class of variants (inframe, splicing, etc.), enrichment in the set of pQTLs without any evidence of RNA mechanism (i.e. no eQTL or transcript isoform QTL) compared to all pQTLs was evaluated by Fisher's exact test, restricted to pQTLs with at least one proxy variant in the gene body (*Figure 3d*).

## Additional information

### Group author details

**HipSci Consortium**
Chukwuma A Agu; **Alex Alderton; Petr Danecek; Rachel Denton; Richard Durbin; Daniel J Gaffney; Angela Goncalves; Reena Halai; Sarah Harper; Christopher M Kirton; Anja Kolb-Kokocinski; Andreas Leha; Shane A McCarthy; Yasin Memari; Minal Patel; Ewan Birney; Francesco Paolo Casale; Laura Clarke; Peter W Harrison; Helena Kilpinen; Ian Streeter; Davide Denovi; Oliver Stegle; Angus I Lamond; Ruta Meleckyte; Natalie Moens; Fiona M Watt; Willem H Ouwehand; Philip Beales**

### Funding

| Funder | Grant reference number | Author |
| --- | --- | --- |
| Wellcome Trust Strategic Award and UK Medical Research Council | WT098503 | Bogdan Andrei Mirauta Dalila Bensaddek Helena Kilpinen |
| Wellcome Trust Strategic Award | 105024/Z/14/Z | Bogdan Andrei Mirauta |
| EMBL Interdisciplinary Postdoctoral (EIPOD) programme under Marie Sklodowska-Curie Actions COFUND | grant number 291772 | Daniel D Seaton Marc Jan Bonder |

The funders had no role in study design, data collection and interpretation, or the decision to submit the work for publication.

### Author contributions

Bogdan Andrei Mirauta, Conceptualization, Data curation, Software, Formal analysis, Validation, Investigation, Visualization, Methodology, Writing - original draft, Writing - review and editing; Daniel D Seaton, Conceptualization, Software, Formal analysis, Validation, Investigation, Visualization, Methodology, Writing - original draft, Writing - review and editing; Dalila Bensaddek, Resources, Data curation, Investigation, Writing - original draft; Alejandro Brenes, Resources, Data curation, Investigation, Writing - original draft, Writing - review and editing; Marc Jan Bonder, Data curation, Software, Investigation, Writing - original draft, Writing - review and editing; Helena Kilpinen, Investigation, Writing - review and editing; HipSci Consortium, Resources, Funding acquisition, Conceptualization; Oliver Stegle, Conceptualization, Formal analysis, Supervision, Funding acquisition, Writing - original draft, Project administration, Writing - review and editing; Angus I Lamond, Conceptualization, Resources, Supervision, Funding acquisition, Writing - original draft, Project administration, Writing - review and editing

### Author ORCIDs

Bogdan Andrei Mirauta (iD) https://orcid.org/0000-0002-9733-292X
Marc Jan Bonder (iD) http://orcid.org/0000-0002-8431-3180
Helena Kilpinen (iD) https://orcid.org/0000-0001-6692-6154

Oliver Stegle https://orcid.org/0000-0002-8818-7193
Angus I Lamond https://orcid.org/0000-0001-6204-6045

### Decision letter and Author response

Decision letter https://doi.org/10.7554/eLife.57390.sa1
Author response https://doi.org/10.7554/eLife.57390.sa2

## Additional files

### Supplementary files

• Supplementary file 1. Comparison of proteome coverage across human proteomics datasets. To facilitate comparison with other datasets we report here the number of proteins and peptides at FDR 1%.

• Supplementary file 2. Disease status. Shown are the number of lines and donors for which matched mRNA and protein data are available.

• Supplementary file 3. Peptides overlapping protein altering variants detected in this study. File containing the list of peptides overlapping protein altering variants or unmapped to the reference genome.

• Transparent reporting form

### Data availability

All data can be accessed via the HipSci data portal, which references EMBL-EBI archives that are used to store the HipSci data. This study includes both cell lines that are consented to be openly accessible as well as cell lines that are subject to managed access, which means a data access application needs to be filed prior to accessing the data.Managed access data from all assays are accessible via EGA under the study EGAS00001001465. Open access genotyping array data and RNA-Seq data are available from ENA under the studies PRJEB11752 and PRJEB7388. Proteomics quantifications (protein group and peptide resolution; MaxQuant output), and run parameters are available on the PRIDE Archive PRIDE (PXD010557). Intermediate result files for this study, such as processed gene expression levels, can be found in Figure 1—source data 2 and 3. Complete summary statistics for the protein and RNA QTL analyses are available at: https://figshare.com/projects/QTL_datasets_for_Population-scale_proteome_variation_in_human_induced_pluripotent_stem_cells_/84233. Analysed data is included in the supplementary files. Scripts used to perform the statistical analyses presented are available at: https://github.com/hipsci/Elife2020 (copy archived at https://github.com/elifesciences-publications/Elife2020).

The following datasets were generated:

| Author(s) | Year | Dataset title | Dataset URL | Database and Identifier |
|---|---|---|---|---|
| Mirauta BA, Seaton DD, Bensaddek D, Brenes MA, Bonder MJ, Kilpinen H, HipSci Consortium, Stegle O, Lamond AI | 2020 | HipSci: the iPSC proteomic compendium | https://www.ebi.ac.uk/pride/archive/projects/PXD010557 | PRIDE, PXD010557 |
| Mirauta BA, Seaton DD, Bensaddek D, Brenes MA, Bonder MJ, Kilpinen H, HipSci Consortium, Stegle O, Lamond AI | 2020 | hipsci_peptide_qtls | https://doi.org/10.6084/m9.figshare.12612800.v4 | figshare, 10.6084/m9.figshare.12612800.v4 |
| Mirauta BA, Seaton DD, Bensaddek D, Brenes MA, Bonder MJ, Kilpinen H, | 2020 | hipsci_transcript_qtls_202_lines | https://doi.org/10.6084/m9.figshare.12608825.v3 | figshare, 10.6084/m9.figshare.12608825.v3 |

| | | | | |
|---|---|---|---|---|
| HipSci Consortium, Stegle O, Lamond AI | | | | |
| Mirauta BA, Seaton DD, Bensaddek D, Brenes MA, Bonder MJ, Kilpinen H, HipSci Consortium, Stegle O, Lamond AI | 2020 | hipsci_eqtls_202_lines | https://doi.org/10.6084/m9.figshare.12608168.v2 | figshare, 10.6084/m9.figshare.12608168.v2 |
| Mirauta BA, Seaton DD, Bensaddek D, Brenes MA, Bonder MJ, Kilpinen H, HipSci Consortium, Stegle O, Lamond AI | 2020 | hipsci_pqtls | https://doi.org/10.6084/m9.figshare.12608027.v2 | figshare , 10.6084/m9.figshare.12608027.v2 |

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
