## [Decision Letter]

**Acceptance summary:**

This exciting study quantifies the genetic architecture of gene expression (eQTL) and protein abundance (pQTL) regulation in iPSCs and will server is an important benchmark for the community. The rigorous comparison of these e/pQTL maps reveals differences that represent expected biological diversity, with a notable example in Figure 3B. Further, this work will serve as a foundation for many novel future studies, including GWAS colocalization and additional omics layers.

**Decision letter after peer review:**

Thank you for submitting your article "Population-scale proteome variation in human induced pluripotent stem cells" for consideration by *eLife*. Your article has been reviewed by three peer reviewers, including Stephen CJ Parker as the Reviewing Editor and Reviewer #1, and the evaluation has been overseen by Patricia Wittkopp as the Senior Editor. The following individuals involved in review of your submission have agreed to reveal their identity: Arushi Varshney (Reviewer #3); Roderic Guigó (Reviewer #4).

The reviewers have discussed the reviews with one another and the Reviewing Editor has drafted this decision to help you prepare a revised submission.

Summary:

Here the authors report pQTL results on 202 iPSC lines from 151 donors which already had RNA-seq data, enabling direct population-scale comparisons of eQTL and pQTL. This manuscript is well-written, clear, and has impactful results. We think it would be a great resource for the field. This is excellent work, and the authors are to be commended on an exciting study. We have a few items of feedback:

Revisions:

1) It is unclear how the acquisition of RNA and proteomics data were related. For example, were the same batches prepped, split, and material frozen for later profiling. Or, were the lines grown up at separate times for each of the different profiling modalities? If the latter, how could this potentially RNA-protein confounding batch effect influence the results?

2) In subsection “pQTL arising from protein-altering variants”, paragraph three, the authors advocate using protein abundance instead of RNA to interpret pathogenic mechanisms of rare variants. However, QTL studies generally have low power to detect effects for SNPs with low MAF. In fact, in this study, the authors used MAF 5% or higher. So, how does one reconcile this assertion to use protein abundance to interpret rare variant effects with the massive sample size it would take to do so. Addressing this idea in the text would be helpful.

3) The authors should make the full summary scan results available so that the rest of the community can use them as a resource.

4) Discussion paragraph three, the word "significant" is missing a "ly" and we'd advise removing that word altogether. In general, it should be used when making a statistical comparison and the associated test and p-value should be provided. This is not mentioned anywhere in the sentence. So, either those results should be disclosed, or a different word without a statistical connotation should be used. The same issue is present in paragraph three of subsection “pQTL arising from protein-altering variants”.

5) The variance component analysis showed effects of culture medium, sex, age etc. Were these taken as covariates in the QTL analysis? Looking at effects of the culture medium, were the culture passage numbers comparable across lines and does that have an effect?

6) For the X chromosome inactivation (XCI) section, it is unclear how exactly the XCI status was quantified. Subsection “RNA and proteome variability” paragraph two and the legend to Figure 1 reference the Materials and methods section but a description of this analysis is missing there. These results are hard to interpret without methodological clarity.

7) Figure 1D – While the random forest model analysis is interesting, the authors should elaborate on their selection of these specific variables. Some other factors that would be informative to include would be MAF and RNA expression level.

8) Figure 2B and 3C – is the grey bar/shaded region in these plots the genomic location of the respective gene? If so, this should be specified in the legend.

9) Figure 3C and the corresponding text briefly describe a pQTL for the VRK2 gene where the pQTL SNP is also a GWAS SNP for schizophrenia risk. The authors should elaborate on the pQTL direction of effect with respect to GWAS risk. Is the variant the lead SNP for GWAS or what is the LD r2 with the lead SNP and do these signals colocalize in that case? Is there some evidence of this protein being relevant in schizophrenia related cellular mechanisms?

10) Figure 4E – This is an interesting example. What is the eQTL/pQTL and trans pQTL direction of effect with respect to the Alzheimer's GWAS risk allele? In this example, the SNP rs1129187 is associated with PEX6 mRNA expression and protein abundance, and also associated with PEX1 and PEX26 abundance. To directly test if the trans-association of the SNP with PEX1 and PEX26 is through the association with PEX6 (complex stability) and not through other mechanisms, have the authors tried to regress out the PEX6 abundance from the association between the SNP and PEX1 or PEX26 and check if the association disappears?

11) Wrong figures are referenced in some places in the manuscript. Figure 3D is referenced before 3B etc.

12) Were there genes for which significant eQTL and also pQTL associations were identified but the variants were independent (low LD r2?)

13) It was confusing that the authors do not clearly distinguish between the variant affecting the phenotype of the gene (transcript or protein expression) and the affected gene. They write "we report 654 genes with a cis pQTL and 3487 genes with a cis eQTL. I assume that will in general find multiple p/eQTLs for a given gene (althought these number do not appear to be reported). These are thus the numbers of p/eGenes, but not of p/eQTLs. However, when they set to investigate replication of p/eQTLs, the numbers correspond to p/eGenes. The authors equate the numbers of QTLs with the number of affected genes. This part could be more clear.

14) Figure 1B. It looks to me that the fraction of the total variance explained by the factors the authors use in their model is much larger for transcriptomics than for proteomics data. I suggest the authors to report this number. If I am correct, this would mean that proteomics data behaves "somehow" more stochastically than transcriptomics data, maybe reflecting technical issues. It maybe also linked to the lack of replication of eQLTs at the protein level.

15) I understand the rationale of using 250Kb for eQTL analysis, since much of the regulation of gene expression is likely to reside in the promoter region. However, I do not see a biological rationale for using the same window for pQTL analysis. I understand that using the same window maybe the only way of making meaningful comparisons between eQTLs and pQTLs, and I think that this is ok. By using the same window the authors are implicitly assessing to what extent variation affecting gene expression also affects protein expression, that is the genetic variation in which the impact on protein expression is mediated by the impact on gene expression. Maybe the authors should acknowledge this.

16) Related to the above. eQTLs tend to cluster around the TSS. Do they observe the same clustering for pQTLs? What is the comparative distribution of p/eQTLs along the tested region?

---

## [Author Response]

Revisions:1) It is unclear how the acquisition of RNA and proteomics data were related. For example, were the same batches prepped, split, and material frozen for later profiling. Or, were the lines grown up at separate times for each of the different profiling modalities? If the latter, how could this potentially RNA-protein confounding batch effect influence the results?

The lines were expanded once for both banking of material and all molecular assays, including RNA and proteomics quantification. Hence, the same batch of cells was split and frozen material was distributed and used for both RNA and proteomics processing. We have clarified this in the main text (Results section), and included further details about the processing steps in the Materials and methods section.

2) In subsection “pQTL arising from protein-altering variants”, paragraph three, the authors advocate using protein abundance instead of RNA to interpret pathogenic mechanisms of rare variants. However, QTL studies generally have low power to detect effects for SNPs with low MAF. In fact, in this study, the authors used MAF 5% or higher. So, how does one reconcile this assertion to use protein abundance to interpret rare variant effects with the massive sample size it would take to do so. Addressing this idea in the text would be helpful.

We agree that this section was not fully clear. We intended to advocate the value of proteome level data in addition to RNA abundance data and to state that our results demonstrate that the proteomics data add another important level of information for annotating specific pathogenic variants. We agree, however, that the sample size is a concern, which limits the frequency spectrum that can be studied using our data (we consider MAF>5% for our analyses). We have reworded implicated sections accordingly.

3) The authors should make the full summary scan results available so that the rest of the community can use them as a resource.

We agree that these data will be useful. We now provide the complete summary statistics for all eQTL, pQTL, transcript QTL and peptide QTL analyses we have conducted. These high-volume data are not suitable for including as supplementary tables and hence have been deposited on figshare. We have referenced them in the Data availability section.

4) Discussion paragraph three, the word "significant" is missing a "ly" and we'd advise removing that word altogether. In general, it should be used when making a statistical comparison and the associated test and p-value should be provided. This is not mentioned anywhere in the sentence. So, either those results should be disclosed, or a different word without a statistical connotation should be used. The same issue is present in paragraph three of subsection “pQTL arising from protein-altering variants”.

We thank the reviewers for highlighting this issue. We have fixed the corresponding sections and avoid statistical connotation in these specific instances, which we feel are not necessary for these specific statements made.

5) The variance component analysis showed effects of culture medium, sex, age etc. Were these taken as covariates in the QTL analysis? Looking at effects of the culture medium, were the culture passage numbers comparable across lines and does that have an effect?

For the QTL analyses we used PEER factors (Stegle et al., 2012) to adjust for batch and other confounding factors. We have experimented with alternative strategies, including to explicitly account for these additional covariates in the analysis. However, the inclusion of PEER factors performed best in terms of maximizing the number of discoveries. In general, we observed that these factors tend to capture broad covariates such as batch or culture media and hence there tends to be no significant benefit of including them separately. Regarding passage number – we agree that this an interesting covariate, and in particular would have added value to the variance component analysis. Unfortunately, the records on passage number were not fully complete for all lines and hence we could not consider this in our study.

6) For the X chromosome inactivation (XCI) section, it is unclear how exactly the XCI status was quantified. Subsection “RNA and proteome variability” paragraph two and the legend to Figure 1 reference the Materials and methods section but a description of this analysis is missing there. These results are hard to interpret without methodological clarity.

We thank the reviewers and apologize for leaving out this detail from the Materials and methods section. We have now added this in the Materials and methods section.

7) Figure 1D – While the random forest model analysis is interesting, the authors should elaborate on their selection of these specific variables. Some other factors that would be informative to include would be MAF and RNA expression level.

We thank the reviewer for these suggestions. We have extended the analysis and the revised Figure 2D now depicts the relevance of MFA and RNA abundance.

8) Figure 2B and 3C – is the grey bar/shaded region in these plots the genomic location of the respective gene? If so, this should be specified in the legend.

The grey box does indeed indicate the genomic location of the respective gene. We have extended the figure caption accordingly.

9) Figure 3C and the corresponding text briefly describe a pQTL for the VRK2 gene where the pQTL SNP is also a GWAS SNP for schizophrenia risk. The authors should elaborate on the pQTL direction of effect with respect to GWAS risk. Is the variant the lead SNP for GWAS or what is the LD r2 with the lead SNP and do these signals colocalize in that case? Is there some evidence of this protein being relevant in schizophrenia related cellular mechanisms?

Thank you for raising this point. The pQTL lead variant is indeed identical to a reported GWAS risk variant for schizophrenia (Yu et al., 2017). Unfortunately, we could not obtain access to the summary statistics, and hence we have not conducted a formal colocalization test for this locus. However, the effect size direction of this pQTL is consistent with prior evidence for the disease relevance of VRK2. Briefly, the risk allele (OR 1.17) is associated with decreased protein expression. Several previous studies have implicated VRK2 in schizophrenia and have linked downregulation of VRK2 RNA abundance to schizophrenia and other neurological disorders (Azimi et al., 2018, Tesli et al., 2016). We have extended the main text accordingly to mention this (subsection “pQTL arising from protein-altering variants”).

10) Figure 4E – This is an interesting example. What is the eQTL/pQTL and trans pQTL direction of effect with respect to the Alzheimer's GWAS risk allele? In this example, the SNP rs1129187 is associated with PEX6 mRNA expression and protein abundance, and also associated with PEX1 and PEX26 abundance. To directly test if the trans-association of the SNP with PEX1 and PEX26 is through the association with PEX6 (complex stability) and not through other mechanisms, have the authors tried to regress out the PEX6 abundance from the association between the SNP and PEX1 or PEX26 and check if the association disappears?

The risk allele at *rs1129187* is associated with increased abundance of both PEX6 RNA and protein level, and is also positively associated with the abundance of the remaining complex subunits PEX26 and PEX1. We have extended the text to include this information.

Regarding the mediating role of PEX6 in the trans association, two lines of evidence indicate that PEX6 is mediating this QTL effect. First, all the complex members are correlated (e.g. r=0.42 for PEX6 and PEX1). Second, we have conducted a regression-based analysis to compare the evidence for a genetic effect on the downstream targets before and after adjusting for PEX6 expression. This results in decreased correlation (r=0.06 vs r=0.29 for PEX1 and r=0.36 versus 0.57 for PEX26), which is consistent with the hypothesized mediation. Nevertheless, this result remains a hypothesis at this stage and hence we have toned down this claim in the main text.

11) Wrong figures are referenced in some places in the manuscript. Figure 3D is referenced before 3B etc.

We thank the reviewer for highlighting these referencing errors. We have carefully revised and checked all references in the paper.

12) Were there genes for which significant eQTL and also pQTL associations were identified but the variants were independent (low LD r2?)

This is indeed a nice addition to our results as presented. In total, we identify 82 genes with significant eQTL and pQTL with independent lead variants (r^2^<0.1). The LD (r^2^) between the reported lead variants is now included in the Figure 1—source data 2, 3. We have added a statement to the main text (subsection “Mapping cis genetic effects on protein abundance”).

13) It was confusing that the authors do not clearly distinguish between the variant affecting the phenotype of the gene (transcript or protein expression) and the affected gene. They write "we report 654 genes with a cis pQTL and 3487 genes with a cis eQTL. I assume that will in general find multiple p/eQTLs for a given gene (althought these number do not appear to be reported). These are thus the numbers of p/eGenes, but not of p/eQTLs. However, when they set to investigate replication of p/eQTLs, the numbers correspond to p/eGenes. The authors equate the numbers of QTLs with the number of affected genes. This part could be more clear.

We have carefully revised the manuscript to clarify the number of eGenes versus distinct eQTL/pQTL. Note that given the moderate sample size of our study, we have limited our analysis to lead e/pQTL variants and hence in most cases there is a one to one mapping of these terms. Nevertheless, we agree that correct terminology is important.

14) Figure 1B. It looks to me that the fraction of the total variance explained by the factors the authors use in their model is much larger for transcriptomics than for proteomics data. I suggest the authors to report this number. If I am correct, this would mean that proteomics data behaves "somehow" more stochastically than transcriptomics data, maybe reflecting technical issues. It maybe also linked to the lack of replication of eQLTs at the protein level.

We agree that these differences are quite striking. The most likely explanation is higher assay noise reflecting the lower sensitivity of quantitative proteomics technologies compared to state-of-the-art deep RNA sequencing. This is indeed also the most likely explanation of the reduced mapping power. We have added a note in the main text (subsection “RNA and proteome variability”) as well as the Discussion section.

15) I understand the rationale of using 250Kb for eQTL analysis, since much of the regulation of gene expression is likely to reside in the promoter region. However, I do not see a biological rationale for using the same window for pQTL analysis. I understand that using the same window maybe the only way of making meaningful comparisons between eQTLs and pQTLs, and I think that this is ok. By using the same window the authors are implicitly assessing to what extent variation affecting gene expression also affects protein expression, that is the genetic variation in which the impact on protein expression is mediated by the impact on gene expression. Maybe the authors should acknowledge this.

We agree that these choices deserve some justification. Besides the expected biological mechanisms, e.g. promoter regulation or proximal enhancers, the choice is primarily motivated by tradeoffs in mapping power to identify genetic effects. Larger testing regions entail a multiple testing burden thereby prohibiting the ability to find eQTL/pQTL variants that are proximal to the TSS. There is no generic recipe for these tradeoffs and our choice can be considered a middle ground when comparing to previous pQTL studies (e.g. 20kb around the gene boundaries in Battle et al., 2015 and 1Mb around the TSS in Sun et al., 2018). We acknowledge that a smaller window may mean that we miss interesting distal effects. We have included a brief justification in the Materials and methods section.

16) Related to the above. eQTLs tend to cluster around the TSS. Do they observe the same clustering for pQTLs? What is the comparative distribution of p/eQTLs along the tested region?

Yes, this locational clustering in the vicinity of the TSS is expected, and indeed prior studies have reported this for eQTL (Kilpinen et al., 2017). The distribution of pQTLs is similar to the one observed for eQTLs. While initially we did not report these results, we follow the reviewer's suggestion and now include them (Figure 2—figure supplement figure 3).